# Altered Immunomodulatory Responses in the CX3CL1/CX3CR1 Axis Mediated by hMSCs in an Early In Vitro SOD1^G93A^ Model of ALS

**DOI:** 10.3390/biomedicines10112916

**Published:** 2022-11-14

**Authors:** Anastasia Sarikidi, Ekaterini Kefalakes, Christine S. Falk, Ruth Esser, Arnold Ganser, Nadine Thau-Habermann, Susanne Petri

**Affiliations:** 1Department of Neurology, Hannover Medical School, 30625 Hannover, Germany; 2Institute of Neuroanatomy and Cell Biology, Hannover Medical School, 30625 Hannover, Germany; 3Institute of Transplant Immunology, Hannover Medical School, 30625 Hannover, Germany; 4Institute of Cellular Therapeutics, Hannover Medical School, 30625 Hannover, Germany; 5Institute of Hematology, Hemostasis, Oncology and Stem Cell Transplantation, Hannover Medical School, 30625 Hannover, Germany

**Keywords:** amyotrophic lateral sclerosis, SOD1^G93A^, motor neurons, human mesenchymal stem/stromal cells, fractalkine

## Abstract

Amyotrophic lateral sclerosis (ALS) is a fatal motor neuron (MN) disease characterized by progressive MN loss and muscular atrophy resulting in rapidly progressive paralysis and respiratory failure. Human mesenchymal stem/stromal cell (hMSC)-based therapy has been suggested to prolong MN survival via secretion of growth factors and modulation of cytokines/chemokines. We investigated the effects of hMSCs and a hMSC-conditioned medium (CM) on Cu/Zn superoxidase dismutase 1^G93A^ (SOD1^G93A^) transgenic primary MNs. We found that co-culture of hMSCs and MNs resulted in slightly higher MN numbers, but did not protect against staurosporine (STS)-induced toxicity, implying marginal direct trophic effects of hMSCs. Aiming to elucidate the crosstalk between hMSCs and MNs in vitro, we found high levels of vascular endothelial growth factor (VEGF) and C-X3-C motif chemokine 1 (CX3CL1) in the hMSC secretome. Co-culture of hMSCs and MNs resulted in altered gene expression of growth factors and cytokines/chemokines in both MNs and hMSCs. hMSCs showed upregulation of CX3CL1 and its receptor CX3CR1 and downregulation of interleukin-1 β (IL1β) and interleukin-8 (IL8) when co-cultured with SOD1^G93A^ MNs. MNs, on the other hand, showed upregulation of growth factors as well as CX3CR1 upon hMSC co-culture. Our results indicate that hMSCs only provide moderate trophic support to MNs by growth factor gene regulation and may mediate anti-inflammatory responses through the CX3CL1/CX3CR1 axis, but also increase expression of pro-inflammatory cytokines, which limits their therapeutic potential.

## 1. Introduction

Amyotrophic lateral sclerosis (ALS) is a rapidly progressive neurodegenerative disease affecting motor neurons (MNs) in the primary motor cortex, brain stem, and spinal cord. Patients suffer from rapidly progressive muscle weakness and atrophy, which ultimately lead to death due to respiratory failure within 2–5 years from symptom onset [1]. Of all ALS cases, 90% are sporadic and only 10% are familial (fALS) [2]. Approximately 20% of all fALS cases are caused by mutations in the gene encoding the Cu/Zn superoxidase dismutase 1 (SOD1) protein, which, under physiological conditions, is a detoxifying enzyme for reactive oxygen species [3,4]. Other ALS mutations have been detected in the TAR-DNA-binding protein 43 (TDP43), fused in sarcoma (FUS), and the recently discovered hexanucleotide repeat expansion of the C9orf72 gene [5,6,7]. Through overexpression of the mutant human SOD1 gene, transgenic animal models can be generated that are clinically and neuropathologically similar to the ALS phenotype [3,8,9].

ALS pathomechanisms include glutamate-mediated excitotoxicity, mitochondrial dysfunction, altered RNA metabolism, disturbed axonal transport, inflammation, oxidative stress, as well as a non-cell autonomous MN death, which is induced by glial cells including astrocytes, microglia, and oligodendrocytes [10]. Due to its multifactorial pathophysiology, no effective therapy has been developed to date. The only approved pharmacological treatments are the glutamate antagonist riluzole, which marginally prolongs patient survival by 3–4 months [11] and, with approval only in the US, Japan, and South Korea, the free radical scavenger edaravone, which has been shown to delay disease progression in a specific subset of ALS patients [12].

Although direct cell replacement seems feasible in some neurological diseases such as Parkinson’s disease, in others, such as ALS, the focus of cell therapeutic approaches is rather on generating a protective environment for degenerating MNs. Human mesenchymal stem/stromal cells (hMSCs) are important therapeutic candidates for a variety of diseases, as they promote regeneration and tissue repair due to their growth-factor rich secretome [13].

Moreover, hMSCs can be isolated from almost all tissues and effectively expanded in vitro [14]. According to the criteria of the International Society for Cellular Therapy, hMSCs are characterized by their capacity for self-renewal, multipotency towards osteogenic, chondrogenic, and adipogenic ontogeny, as well as the expression of a cluster of differentiation, including CD73, CD90, and CD105 (while lacking CD14, CD34, CD45, and human leukocyte antigen-DR) [15]. Bone marrow is the best characterized source of hMSCs. Promising alternatives are umbilical blood cord, adipose tissue, placenta, and others [16]. hMSCs can differentiate into all mesenchymal lineages, including bone, adipose, cartilage, muscle, and myelosupportive stroma cells [17].

More important to their therapeutic potential, however, are their low immunogenicity and immunomodulatory effects [18]. hMSCs are able to switch between pro- and anti-inflammatory phenotypes and induce or suppress immune effector cells. They can release a variety of immunomodulatory cytokines/chemokines and trophic factors with neurogenic properties [19]. In a previous study of the immortalized NSC-34 cell line, which is a hybrid cell line derived from neuroblastoma cells and MNs from mouse embryonic spinal cords, we previously showed that murine MSC and a murine mesenchymal stem/stromal cell-conditioned medium (MSC-CM) were able to regulate glial responsiveness via the CX3CL1/CX3CR1 axis [20]. The CX3CL1/CX3CR1 axis is known to maintain microglia-mediated autophagy. Impairment of this process enhances neurodegeneration [21]. CX3CL1 (fractalkine) itself is the only chemokine expressed by neurons, also by MNs, and has been shown to increase neuroprotective capacities in microglia via interaction with its receptor CX3CR1 [22]. As hMSCs are able to excrete CX3CL1 and may provide an anti-inflammatory environment for MNs, we aimed to further elucidate whether regulation of CX3CL1/CX3CR1 also occurred in primary MNs of the SOD1^G93A^ mouse model of ALS. Apart from characterizing CX3CL1/CX3CR1 in the crosstalk of MNs and hMSCs, we tried to elucidate whether changes in this axis coincided with altered gene expression of growth factors and cytokines/chemokines in MNs and/or hMSCs or only arose during co-culture of both cell types. We also aimed to characterize the hMSC secretome and potential pro- or anti-inflammatory changes in transgenic MNs cultured in the presence of hMSC-CM or hMSCs.

## 2. Materials and Methods

### 2.1. Ethics Statement

All animal experiments were conducted in strict accordance with the German Animal Welfare Law and were by the Lower Saxony State Office for Consumer Protection and Food Safety (LAVES, §4 notification according to the German Animal Welfare Act).

### 2.2. Animals

All transgenic (SOD1^G93A^) and wildtype mice were housed under controlled conditions with a light-dark cycle of 12:12 h in groups of a maximum of 6 animals (Makrolon cage type II; Uno, Zevenaar, The Netherlands). Animals had free access to food and water at all times. Mice overexpressing the human SOD1^G93A^ gene in a high copy number were purchased from Jackson Laboratories (stock number 004435; Bar Harbour, ME, USA). Hemizygous transgenic male SOD1^G93A^ mice were mated with wildtype C57BL/6 females that were either purchased from Jackson Laboratories (stock number 000664; Bar Harbour, ME, USA) or obtained by own breeding.

### 2.3. Human Mesenchymal Stromal Cells

#### 2.3.1. Isolation of hMSCs

Human MSC isolation was performed after written consent from the bone marrow of healthy donors (*n* = 5). The production and use of hMSCs has been approved by the Ethics Committee of Hannover Medical School, Germany (approval no. 2858–2015).

The puncture was performed after local anesthesia with a bone marrow puncture needle 15 G (Almond & Rupp, Erkrath, Germany) at the pelvic crest. Approximately 10–15 mL of bone marrow was aspirated. Human bone marrow was diluted with a ratio of 1:4 with sterile phosphate-buffered saline (PBS) and the cell count was determined. Cells were cultivated with a concentration of 1–2 × 10^6^ cells/cm^2^ with 50 mL CellGro^®^ MSC medium (Cellgenix, Freiburg im Bresgau, Germany) and enriched with 1% Penicillin/Streptomycin, heparin 2 IU/mL, and 8% human plate lysate in 175 cm^2^ tissue culture flasks (Sarstedt, Nümbrecht, Germany). After four days, cells were washed thrice with 30 mL PBS to avoid blood residues. The medium was changed every 4–5 days and adherent cell growth was microscopically controlled. Cells were subcultured until they reached 80% of confluence and passaged at a density of 2 × 10^3^ cells/cm^2^. The remaining cells were cryopreserved at a density of 1 × 10^6^ cells/mL in MSC medium with 5% dimethylsulfoxide (DMSO, Sigma Aldrich, Taufkirchen, Germany) and frozen at 70 °C. Cells were used for culture at a low passage (≤4) [23,24].

#### 2.3.2. Differentiation of hMSCs

According to the recommendation of the International Society for Cellular Therapy, expression of hMSC surface markers was confirmed by single platform fluorescence-activated cell sorting (FACS) and its capacity to differentiate into chondrocytes, osteoblasts, and adipocytes was analyzed and validated in multiple independent experiments. In order to meet the criteria, collected cells were analyzed for their expression of CD73, CD90, and CD105, as well as the absence of CD3, CD14, CD19, CD34, CD45, and HLA-DR expression in multiple independent experiments [15].

For identification of MSCs, as well as the determination of HLA-DR+ and CD45 expression, the fluorochrome-conjugated monoclonal antibodies (mAb) CD90FITC, HLA-DR PB, CD45PC7 (Beckman Coulter, Krefeld, Germany) and CD73PE, CD105APC (Becton Dickinson, Heidelberg, Germany) were used. For cell composition, the fluorochrome-conjugated monoclonal antibodies (mAb) CD90FITC, CD34PE, CD19PC7, CD3APC, CD14 PB, and CD45KO (all Beckman Coulter, Krefeld, Germany) were used. After 15 min of incubation at ambient temperature in the dark, erythrocyte lysis was carried out by adding NH4Cl solution if necessary. Single platform analysis using flow-count fluorospheres on a Navios™ 10 color flow cytometer (Beckman Coulter, Krefeld, Germany) was performed. To distinguish between viable and dead cells, 7-Amino-Actinomycin-D (7-AAD) was used.

The average percentage of cells expressing CD73, CD90, and CD105 in the population of bone marrow hMSCs was 97.5% (calculated by five distinct production processes of five donors, listed in Appendix A).

In addition, the capacity to differentiate into chondrocytes, osteoblasts, and adipocytes was evaluated [15]: For chondrocyte differentiation, hMSCs were cultivated in tubes for 21 days in Stem MACS Chondrodiff medium (Miltenyi Biotec, Gladbach, Germany), enriched with 1% Penicillin/Streptomycin. A medium change was performed every 3–4 days. After four weeks, cell pellets were fixed with 4% paraformaldehyde (PFA), washed twice with PBS, embedded in Tissue-Tek with O.C.T. compound (Sakura Finetek, Torrance, CA, USA) and fast-frozen in liquid nitrogen. By using a microtome, the sample was cryosectioned with a thickness of 12 µm. For cartilage-specific proteoglycan examination, sections were stained with Alcian blue 8GX (Sigma Aldrich, Taufkirchen, Germany) and counterstained with nuclear fast red (Sigma Aldrich, Taufkirchen, Germany).

For osteogenic differentiation, hMSCs were stimulated for 21 days in STEM Macs Osteodiff medium (Miltenyi Biotec, Gladbach, Germany), enriched with 1% Penicillin/Streptomycin. A medium change was performed every 3–4 days. After four weeks cells were washed once with 1xPBS and fixed with 8% PFA. Afterward, cells were washed trice with distilled water and stained with Alizarin Red solution (Sigma Aldrich, Taufkirchen, Germany) to detect calcium-rich deposits and SIGMA FAST BCIP/NBT (Sigma Aldrich, Taufkirchen, Germany) to identify alkaline phosphatase activity.

For adipocyte differentiation, the MSC medium was replaced with the StemPro Adipocyte Differentiation Basal medium (Thermo Fisher, Schwerte, Germany) and enriched with 1% Penicillin/Streptomycin for 14 days. After two weeks, cells were washed twice with 1× PBS and fixed with 4% PFA. Cells were stained with Oil Red O (Sigma Aldrich, Taufkirchen, Germany) to enable detection of lipid vacuoles.

### 2.4. Human MSC-Conditioned Medium

hMSCs were cultured as described above and seeded at a density of 1 × 10^5^ cells/well in 6-well plates (Thermo Fisher Scientific, Schwerte, Germany). After rinsing with 1xPSB, the Neurobasal MN culture medium (Gibco, Darmstadt, Germany) containing 5% horse serum (Pan Biotech, Aidenbach, Germany), 2% B27 supplement (Invitrogen, Darmstadt, Germany), and 10% Glutamax (Invitrogen) was added. After 24 h, the supernatant was retrieved, centrifuged at 2000× *g* rpm for 5 min, and rinsed through a 0.22 µm filter to remove cell debris and remaining cells. The obtained conditioned medium was stored at −70 °C until further use.

### 2.5. Primary Motor Neuron Culture

Primary MN were obtained by euthanizing pregnant wildtype female animals that were mated with SOD1^G93A^ males. Embryos (day 12.5) were euthanized as well and the lumbar spinal cords of the embryos were dissected. Enrichment of MN was conducted, as previously described [25], by immunopanning using p75^NTR^ (Abcam, Cambridge, UK). Eight-well chamber slides (VWR International GmbH, Hannover, Germany) or 6-well plates (Thermo Fisher Scientific, Schwerte, Germany) were precoated with poly-L-ornithin (Sigma, Taufkirchen, Germany), 1:1000 in borate buffer (0.15 M, pH 8.35) for 30 min at 37 °C. The diluted poly-L-ornithin was removed and slides/wells were washed twice with distilled water. After washing, slides/wells were coated with laminin (Invitrogen, Darmstadt, Germany), diluted 1:100 in Hanks Balanced Salt Solution (HBSS; Invitrogen) at room temperature (RT) until cell harvest. For cultivation, Neurobasal motor neuron culture medium (Gibco, Darmstadt, Germany) containing 5% horse serum (Pan Biotech, Aidenbach, Germany), 2% B27 supplement (Invitrogen), and 10% Glutamax (Invitrogen) was used. The horse serum used in the neurobasal medium was previously heat inactivated for 45 min at 56 °C. MN were either plated in neurobasal hMSC-conditioned medium or in an equal amount of hMSCs, dependent on experimental design.

For quantification of MN numbers, three independent experiments were conducted at different timepoints. Two technical replicates per genotype were seeded on two distinct wells. Means were calculated out of the technical replicates and according to the biological replicates (wildtype or SOD1^G93A^) of each experiment. Means were then calculated out of the three independent experiments (according to their genotype) and expressed as means with SEM per genotype.

### 2.6. Staurosporine (STS)-Induced Toxicity

STS is a non-selective protein kinase inhibitor that induces apoptosis in various cell types. After 5 days in vitro (DIV), cells were incubated for 24 h with different concentrations of STS (0.2, 0.5, 1, 3, and 5 µM, respectively) (Sigma, Taufkirchen, Germany), dissolved in 1 mM DMSO to test for concentration-dependent neurotoxic effects by immunocytochemistry and LDH assay. The concentration of 0.5 µM was the first concentration showing significant reductions in MN number after 24 h. Further, increasing STS concentrations enhanced this effect; thus, 0.5 μM of STS was used for the main experiments. As a control, cells not exposed to STS were used.

### 2.7. Lactate Dehydrogenase (LDH) Cytotoxicity Assay

LDH is a soluble cytosolic enzyme present in most eukaryotic cells and is released into the culture medium due to damage to the plasma membrane in the case of cell death. The increase in the LDH activity in culture supernatant is proportional to the number of lysed cells. In our experiment, the LDH concentration was measured using the Pierce LDH assay kit, according to manufacturer’s instructions (Thermo Fischer Scientific, Damstedt, Germany), after 5 DIV and after 24 h of STS exposure. Seventy microliters of supernatant were collected und centrifuged to avoid cell debris, 50 µL of which were incubated at RT with 50 µL of reaction solution. Stop solution was added after a 30 min incubation period and LDH levels were determined at 492/630 nm.

### 2.8. Immunocytochemistry

On the 5th DIV, cells were incubated for 1 h with the MitoTracker stain (MitoTracker™ Red CMXRos) mixed with neurobasal medium 1:100 nM und washed once with 1× PBS. Cells were fixed in 4% paraformaldehyde for 20 min at RT. Afterward, wells were washed thrice with 1× PBS. Cells were blocked in blocking buffer consisting of 0.2% Triton-X-100 (Sigma), 10% goat serum (Invitrogen), and 2.5% BSA at RT for 30 min. After blocking, primary antibodies were added (rabbit anti-islet1 [1:500; Abcam]) in blocking solution and incubated overnight at 4 °C. The next day, wells were washed thrice with 1× PBS. Secondary antibodies (AlexaFluor 555 goat anti-rabbit [1:1000; Invitrogen]) were diluted in blocking buffer and incubated for 2 h at RT. Cells were washed thrice with 1× PBS and once with distilled water. After washing was completed, cells were mounted in mounting medium (Mowiol; Roth, Karlsruhe, Germany) containing 0.1% DAPI (Sigma), and visualization took place through fluorescence microscopy (BX61; Olympus Deutschland GmbH, Hamburg, Germany). Image acquisition was performed with Cell F and CellSens Dimensions Ink. Software 1.18 (Build 16686, Olympus Deutschland GmbH, Hamburg, Germany), using a 20× objective and a numerical aperture of 0.17.

### 2.9. Multiplex Analysis of Cytokines and Chemokines

hMSCs obtained from 5 healthy donors were cultured with a density of 1 × 10^5^ cells/well on 6-well plates (Thermo Fisher Scientific, Schwerte, Germany) in CellGro^®^ MSC medium (Cellgenix, Freiburg im Bresgau, Germany) enriched with 1% Penicillin/Streptomycin, heparin 2 IU/mL, and 8% human plate lysate. After 5 min (indicated as timepoint 0), 24 and 48 h cell supernatant was collected. As a control, pure MSC medium was used. Media were measured using multiplex technology (Bio-Plex System, Hercules, USA) (Merck: MILLIPLEX MAP Human Cytokine/Chemokine Magnetic Bead Panel, MILLIPLEX MAP Human Pituitary Magnetic Bead Panel 1 and MILLIPLEX MAP Human Neurodegenerative Disease Magnetic Bead Panel 3 (Catalogue number: HCYTOMAG-60K, HPTP1MAG-66K, HNDG3MAG-36K)). Pure MSC medium (1% Penicillin/Streptomycin, heparin 2 IU/mL, and 8% human plate lysate) was used as a negative control, and thus subtracted from all measured values. Supernatants (25 µL) were co-incubated with 25 µL of the missed beads, and 25 µL assay buffer for 45 min at RT in the dark on a plate shaker. After three washing steps, 25 µL mixed biotinylated secondary antibodies were added and incubated for 45 min, followed by three washing steps. Quantification was performed by adding SA-PE (1:100 dilution in assay buffer) for 30 min in the dark. Individual standard curves and concentrations for each analyte were calculated by the Bio-Plex Manager 6.2 software (Bio-Rad, Hercules, USA); the detection range for each analyte was between 1 pg/mL as the lowest and 10 µg/mL as the highest concentrations. Pure MSC medium was used as a negative control, and thus subtracted from all measured values.

### 2.10. Quantitative Real-Time PCR (qRT-PCR)

MN in mono-culture and co-cultured with hMSCs were lysed after 5 DIV in order to perform gene expression analysis by qRT-PCR. The medium was removed, cells were washed with 1xPBS, and cells were lysed using the Rneasy Micro Kit (Qiagen, Venlo, The Netherlands), according to manufacturer’s instructions. A total quantity of 1 and 0.2 ng cellular RNA was reverse transcribed to complementary DNA (cDNA) with the QuantiTect Reverse transcription kit (Qiagen, Venlo, The Netherlands), according to the manufacturer’s protocol. qRT-PCR experiments were performed using cDNA from 5, 20, or 50 ng of total RNA from at least four independent experiments. An amount of 1.75 µM forward/reverse primer and Power SYBR-Green PCR Master Mix (Life Technologies, Darmstadt, Germany) or appropriate Assay Mix (Applied Biosystems, Darmstadt, Germany), combined with TaqMan Universal Master Mix (2×) (Applied Biosystems, Darmstadt, Germany), were applied. Standard cycling conditions were applied: 90 °C/10 min. and 40 cycles at 95 °C/15 s and 60 °C/1 min. StepOne instrument and software (Applied Biosystems, Waltham, MA, USA) were used (threshold was set at 0.2). For quantitative analysis, the following reference genes: hypoxanthine phosphoribosyltransferase 1 (HPRT1), peptidylprolyl isomerase A (Ppia), glycerinaldehyd-3-phosphat-dehydrogenase (GAPDH), β-actin and tubulin, and β2-mikroglobulin (b2m), respectively, were applied and expression patterns were analyzed by comparative Ct method (2^−ΔΔ^Ct) [26].

For qRT-PCR, specific human and mouse primers were used for analysis of gene expression of hMSCs and mouse primary MNs, respectively. Human- and murine-specific primer pairs were tested with corresponding human and mouse cDNA to confirm the correct length of PCR products via gel electrophoresis on 4% agarose gels. PCR reactions with human primer pairs and murine cDNA or murine primer pairs and human cDNA did not yield PCR products as assessed by gel electrophoresis, thereby confirming species-specificity. Primers are listed in Appendix A.

### 2.11. Statistical Analysis

Data were expressed as means ± SEM standard error of the mean of at least three independent pooled experiments. Statistical analysis was performed using GraphPad Prism version 9 (GraphPad Software Inc., San Diego, CA, USA). For the comparison of two groups, an unpaired *t*-test was used, and for the comparison of three or more groups, a one-way ANOVA and Tukey’s multiple comparisons post-hoc test was applied. For two groups with more than two parameters, a two-way ANOVA and Tukey’s multiple comparisons post-hoc test was used. Differences were considered significant in cases when the *p*-value < 0.05 and is represented as follows: *: *p*-value < 0.05, **: *p*-value < 0.01, and ***: *p*-value < 0.001.

## 3. Results

### 3.1. Effects of hMSCs and hMSC-CM on MNs In Vitro

#### 3.1.1. Beneficial Effects of hMSCs on MN Survival in hMSC-MN Co-Cultures

We previously showed that murine MSC-CM exerted neuroprotective effects on MNs due to the release of growth factors and cytokines/chemokines of hMSCs in vitro [20] and that repeated intraspinal injections of hMSCs resulted in improved motor performance of SOD1^G93A^ mice in vivo [27]. To further elucidate the impact of hMSCs and hMSC-CM on MNs, SOD1^G93A^ and wildtype MNs were either cultured in monoculture (MN monoculture), co-cultured on hMSCs (MN-hMSC co-cultures), or incubated in hMSC for 5 DIV. MN number was determined by the early MN marker islet1 combined with a marker for mitochondrial activity (MitoTracker). Double positive MNs were counted. Exemplary stainings for all conditions (SOD1^G93A^ or wildtype MN monocultures, MN-hMSC co-cultures, and MN cultured in hMSC-CM) can be seen in Figure 1a–f.

There was a significant increase in MitoTracker^+^-islet1^+^-MNs when MNs were co-cultured with hMSCs (MNs-hMSCs) compared with MN monocultures and MNs incubated in hMSC-CM. This increase was similar for both wildtype and SOD1^G93A^ MNs (Figure 2).

#### 3.1.2. Slight but Not Significantly Increased MN Survival after STS-Induced Apoptosis in MN-hMSC Co-Cultures

Based on the observation of increased numbers of MNs in co-culture with hMSCs, we aimed to examine whether hMSCs could rescue MNs under stress conditions. As ALS usually manifests itself in late adulthood, embryonic MNs do not reflect symptomatic disease stages. However, pathologic phenotypes can be mimicked by stress-induced MN cytotoxicity. A commonly used neuronal stressor is STS [28,29]. First, an effective STS concentration inducing MN death was established. To test the concentration-dependent neurotoxic effects induced by STS, MNs co-cultured with hMSCs were incubated for 24 h in a concentration range between 0.2 to 5 µM STS (Appendix A). Double positive islet1^+^-MitoTracker^+^ MNs were determined. Shorter time points (1 h) did not induce MN toxicity and longer ones (48 h) resulted in total MN apoptosis (data not shown). The first significant reduction in MN number was achieved after 24 h, with a concentration of 0.5 µM STS for both wildtype and SOD1^G93A^ MNs when co-cultured on hMSCs. Higher STS concentrations resulted in significantly higher toxic effects. Results were also confirmed by LDH assay, as the first significant increase in LDH levels was detected at a concentration of 0.5 µM STS after 24 h of STS exposure (Appendix A). Higher STS concentrations were significantly toxic for both MNs and hMSCs. Thus, a concentration of 0.5 μΜ STS was used for further examination of potential neuroprotective effects on MNs exerted from hMSCs and hMSC-CM.

There was a tendency toward lower numbers of SOD1^G93A^ MNs compared with wildtype MNs in all conditions (monoculture, hMSC-CM and co-culture) when exposed to 0.5 μM STS; however, the level of significance was not reached. Wildtype and SOD1^G93A^ MN numbers were slightly increased by hMSC-CM or hMSC co-culture, but no significant neuroprotective effects against STS-neurotoxicity were detected (Figure 3).

### 3.2. Growth Factors and the Chemokine CX3CL1 in the hMSC Secretome

#### 3.2.1. Detection of VEGF, BDNF, CNTF, FGF-2, PDGF, and CX3CL1 in the hMSC Secretome

To examine the mechanisms underlying hMSC-mediated effects, the levels of growth factors and chemokine CX3CL1 excreted in the hMSC medium were measured by multiplex analysis (Figure 4a–g). Levels of all investigated growth factors (FGF-2, BDNF, CNTF, PDGF-AA, and PDGF-AB) decreased over time (24 and 48 h). For BDNF, CNTF, and PDGF-AB, this decrease was significant after both 24 and 48 h (Figure 4b,c,f). Only VEGF showed a significant increase in the cell medium over time (both after 24 and 48 h) (Figure 4a). The chemokine CX3CL1 significantly increased after 24 h and a significant decrease was registered after 48 h (Figure 4g).

#### 3.2.2. Differential Alterations in the Gene Expression of CX3CL1 and Its Receptor CX3CR1 with the Interleukins IL1b and IL8 in hMSCs

To further examine potential alterations in hMSC gene expression of growth factors and CX3CL1 hMSCs induced by wildtype versus transgenic MN co-culture, gene expression of growth factors and chemokines/cytokines was specifically analyzed in hMSCs by human-specific primers (Figure 5 and Figure 6).

There were no differences in hMSC mRNA expression of FGF-2, NGF, CNTF, BDNF, GDNF, VEGF, and IGF1 when hMSCs were co-cultured with either SOD1^G93A^ or wildtype MNs (Figure 5a–h).

mRNA levels of the chemokine CX3CL1 and its receptor CX3CR1 were significantly upregulated in hMSCs co-cultured with SOD1^G93A^ MNs compared with hMSC co-cultured with wildtype MNs (Figure 6a,b). mRNA transcription of IL1b was significantly downregulated in hMSCs when cultured with SOD1^G93A^ MNs compared with hMSCs co-cultured with wildtype MNs (Figure 6c). There was no difference in mRNA transcription of IL6 in hMSCs co-cultured with SOD1^G93A^ MNs compared with hMSCs co-cultured with wildtype MNs (Figure 6d). IL8 gene expression was significantly downregulated in hMSCs co-cultured with SOD1^G93A^ MNs compared with hMSCs co-cultured with wildtype MNs (Figure 6e).

### 3.3. 1 hMSC-Dependent Changes in Growth Factor and Chemokine/Cytokine Gene Expression of Wildtype and SOD1^G93A^ MNs

In order to assess the impact of neurotrophins and chemokines released by hMSCs on MN gene expression, mRNA levels of growth factors and chemokines/cytokines of MNs cultured on hMSCs were determined via mouse specific primers (Figure 7 and Figure 8). CNTF gene expression was significantly downregulated in both wildtype- and SOD1^G93A^ MNs co-cultured with hMSCs compared with wildtype and SOD1^G93A^ MN monocultures (Figure 7a). NGF gene expression was significantly downregulated in both wildtype- and SOD1^G93A^ MNs co-cultured on hMSCs compared with wildtype and SOD1^G93A^ MN monocultures (Figure 7b). BDNF gene expression was significantly upregulated in both wildtype and SOD1^G93A^ MNs co-cultured on hMSCs compared with wildtype and SOD1^G93A^ MN monocultures (Figure 7c). mRNA transcription of GDNF was significantly upregulated in SOD1^G93A^ MNs co-cultured on hMSCs compared with SOD1^G93A^ MN monocultures (Figure 7d). FGF-2 gene expression was unaltered in both wildtype and SOD1^G93A^ MNs co-cultured on hMSCs compared with wildtype and SOD1^G93A^ MNs (Figure 7e). VEGF gene transcription was not regulated in either wildtype or SOD1^G93A^ MNs co-cultured on hMSCs compared with wildtype or SOD1^G93A^ MN monocultures (Figure 7f). IGF1 gene expression was significantly downregulated in SOD1^G93A^ MNs co-cultured on hMSCs compared with SOD1^G93A^ MN monocultures (Figure 7g). IGF2 gene expression was not altered in either wildtype or SOD1^G93A^ MN mono or co-cultures compared with wildtype or SOD1^G93A^ MN mono or co-cultures (Figure 7h).

Gene expression of the chemokine CX3CL1 showed a significant downregulation in both wildtype and SOD1^G93A^ MNs co-cultured on hMSCs compared with wildtype and SOD1^G93A^ MN monocultures (Figure 8a). Conversely, its receptor CX3CR1 was significantly upregulated in both wildtype and SOD1^G93A^ MNs co-cultured on hMSCs compared with wildtype and SOD1^G93A^ MN monocultures (Figure 8b). Gene expression of the pro-inflammatory cytokine TNFα showed a significant upregulation in SOD1^G93A^ MNs co-cultured on hMSCs compared with both wildtype and SOD1^G93A^ MN monocultures. Furthermore, a significant TNFα upregulation in SOD1^G93A^ MNs co-cultured on hMSCs was detected compared with wildtype MNs co-cultured on hMSCs (Figure 8c). mRNA transcription of IL6 and IL8 was significantly upregulated in SOD1^G93A^ MNs co-cultured on hMSCs compared with wildtype and SOD1^G93A^ MN monocultures and wildtype MNs co-cultured on hMSCs (Figure 8d,e).

## 4. Discussion

Over the past several years, cell therapy has been explored as an alternative to pharmacological treatment in ALS to prevent MN degeneration by providing a trophic environment. hMSCs isolated from healthy donors are a promising source of neuroprotective growth factors and immunomodulatory chemokines/cytokines for degenerating MNs [30]. We demonstrated in a previous study that MSCs provided trophic support for MNs and NSC-34 cells in vitro and that MSC-CM was neuroprotective for MNs by releasing growth factors and cytokines/chemokines [20]. In the present study, we aimed to characterize the impact of hMSCs and hMSC-CM on primary wildtype and mutant SOD1^G93A^ MNs as an early model of ALS and focus more specifically on the hMSC secretome and gene expression changes induced in both wildtype and transgenic MNs and in hMSCs by co-culture of both cell types.

The main findings of the present study include: (a) A higher viability of MNs in co-culture with hMSCs, but no significant neuroprotection against STS toxicity; (b) Detection of VEGF, BDNF, CNTF, FGF-2, PDGF, and CX3CL1 in the secretome of hMSCs; (c) Upregulation of anti-inflammatory chemokines (CX3CL1 and its receptor CX3CR1) and downregulation of pro-inflammatory cytokines (IL1b and IL8) in hMSCs when co-cultured with SOD1^G93A^ MNs; (d) Modification of MN gene expression of neuroprotective growth factors (CNTF, NGF, and BDNF) in both wildtype and SOD1^G93A^ MNs, and specific upregulation of GDNF only in SOD1^G93A^ MNs, as well as upregulation of the anti-inflammatory chemokine receptor CX3CR1 in both wildtype and SOD1^G93A^ MNs by hMSC co-culture; (e) Upregulation of gene expression of the pro-inflammatory cytokines TNFα, IL6, and IL8 in MNs (with higher levels in SOD1^G93A^ MNs) by hMSC co-culture.

Slightly higher MN viability was registered when MNs were co-cultured on hMSCs. Exposure to STS reduced MN numbers in all conditions, but no significant neuroprotective effect of hMSCs against STS-toxicity could be detected. Vulnerability of SOD1^G93A^ MNs was not significantly higher compared to wildtype MNs, suggesting that longer culture periods may be necessary to mimic ALS onset in this early in vitro model.

Several in vivo studies of the SOD1^G93A^ mouse model have shown neuroprotective effects of hMSCs [31,32,33], but hMSCs only marginally prolonged survival in clinical trials of ALS patients [34,35,36,37].

To elucidate the controversial findings between in vivo studies in mouse models and clinical trials, studies characterizing hMSCs demonstrated that the beneficial effects of hMSCs were mostly exerted by their secretome [38,39]. Thus, one of the goals of the present study was to characterize the secretome of hMSCs and elucidate whether specific hMSC-induced gene expression changes occur in an ALS genotype (SOD1^G93A^ MNs). Examination of the secretome of hMSCs revealed secretion of BDNF, CNTF, and PDGF-AB, decreasing over time and sustained detection of VEGF, whereas levels of CX3CL1 showed a non-sustained increase. Gene expression of both CX3CL1 and its receptor CX3CR1 were significantly upregulated in hMSCs when co-cultured with SOD1^G93A^ MNs. Contrarily, in MNs co-cultured with hMSCs, CX3CL1 was significantly downregulated, whereas its receptor was significantly upregulated. Interestingly, this effect was not SOD1^G93A^-dependent. Upregulation of CX3CR1 in MNs in hMSC co-culture could be interpreted as a reaction to the increased secretion and gene expression of CX3CL1 by hMSCs.

CX3CL1 (fractalkine) is constitutively expressed in neuronal cells and exerts multiple functions via its CX3CR1 receptor, which is primarily located on microglia, as well as in monocytes, dendritic cells, natural killer cells, and T cells [40,41]. CX3CL1 exists in both membrane-bound and soluble forms, and therefore, differs from other chemokines, which are almost exclusively secreted [42]. hMSCs can express and excrete CX3CL1, and thus shape microglial immune responsiveness. Several groups, including ours, have already shown that hMSCs exert beneficial effects on activated microglia in vitro by altering their phenotype and reactivity toward a more neuroprotective phenotype via the release of CX3CL1 and its interaction with the microglial CX3CR1 receptor [19,20,43]. Cardona et al. demonstrated that CX3CR1 deficiency in transgenic mouse models of neurodegenerative diseases led to dysregulated and neurotoxic microglial responses [40]. In ALS pathogenesis, microglial neurotoxicity strongly depends on microglial activation and phenotypes as late stage M1 microglia are described to favor inflammation and consecutive MN loss, whereas M2 microglia, seen at early stages of ALS, protect MNs from degeneration by excreting anti-inflammatory cytokines [44]. Apart from immunomodulatory responses, CX3CL1 and CX3CR1 can also mediate migration of hMSCs to the site of injury in the CNS, and therefore activate regenerative pathways [45].

Neuroinflammation is one of the main pathomechanisms in ALS [10]. It refers to an innate immune response mediated by “activated” glial cells, astrocytes and microglia, and to a lesser extent, by the circulating immune cells (monocytes, neutrophils, and lymphocytes) [46,47,48]. In ALS and especially the SOD1^G93A^ mouse model, neuroinflammation is already observed at early stages and becomes more pronounced with greater disease severity and progression [49,50]. During the systemic inflammation at later disease stages, inflammatory markers are dysregulated and coincide with faster disease progression [51]. IL1b, IL6, IL8, and TNFα are key players in ALS inflammatory pathogenesis as they are correlated with glial activation [52,53]. In line with this, inhibition or deletion of IL1b (or its receptor) or TNFα results in increased MN viability and prolonged survival of ALS mice [54,55]. Within this study, we found that two major inflammatory cytokines, IL1b and IL8, were significantly downregulated in hMSCs when they were co-cultured with SOD1^G93A^ MNs as opposed to co-cultured with wildtype MNs, implying immunomodulatory cross-talk. On the contrary, IL8, together with IL6 and TNFα, were significantly upregulated in SOD1^G93A^ MNs co-cultured with hMSCs compared with wildtype MNs co-cultured with hMSCs, suggesting that hMSCs do not alleviate ALS-related inflammatory mechanisms within this experimental context.

hMSCs secretome analysis revealed a sustained increase in VEGF, a growth factor best known for its angiogenic properties whose protective effects on degenerating MNs have been shown in vitro and in vivo [56,57,58]. Gene expression of VEGF and other growth factors such as BDNF, GDNF, CNTF, and NGF was also altered in MNs in the presence of hMSCs. Whereas BDNF, CNTF, and NGF were upregulated in both wildtype and SOD1^G93A^ MNs co-cultured with hMSCs, GDNF showed a specific upregulation only in SOD1^G93A^ MNs when co-cultured with hMSCs. All of these are well-characterized trophic factors for the central nervous system [59]. Growth factors support physiological processes, such as neural differentiation, synapse maintenance, axonal growth, survival, and neurogenesis [60,61]. As the observed alterations in gene expression of BDNF, CNTF, and NGF were not dependent on the SOD1^G93A^ genotype, the observed increase in the presence of hMSCs does not seem to reflect ALS-specific regulations. In contrast, GDNF, which was only upregulated in SOD1^G93A^, implies an ALS genotype-dependent gene regulation associated to the presence of hMSCs.

Multiple studies have shown beneficial effects of growth factors on MNs and in ALS; however, outcomes of clinical trials using growth factors have only marginally prolonged life expectancy of patients so far [62]. Although growth factors are generally known for their neuroprotective properties, the roles of FGF-2 and NGF have been controversial, especially when acting in a non-cell autonomous manner [63,64,65].

To conclude, our in vitro analyses provide evidence that hMSCs can provide a beneficial environment for MNs mainly via growth factor secretion and stimulation of growth factor gene expression. These cells modulate MN immune responsiveness and possibly microglial activation through CX3CL1/CX3CR1, but also seem to increase gene expression of pro-inflammatory cytokines in SOD1^G93A^ ALS motor neurons, which may limit their therapeutic potential.

## 5. Conclusions

(a)Higher viability of both wildtype and SOD1^G93A^ MNs in co-culture with hMSCs, but no significant protection against STS-induced apoptosis;(b)Regulation of the inflammatory CX3CL1/CX3CR1 axis in SOD1^G93A^ MN-hMSC co-culture;(c)Increased gene expression of GDNF and pro-inflammatory cytokines IL6, IL8, and TNFα in SOD1^G93A^ MNs co-cultured with hMSCs;(d)Potential limitations of the therapeutic efficacy of hMSCs in ALS due to cellular crosstalk resulting in increased pro-inflammatory rather than anti-inflammatory signaling.

## Figures and Tables

**Figure 1 biomedicines-10-02916-f001:**
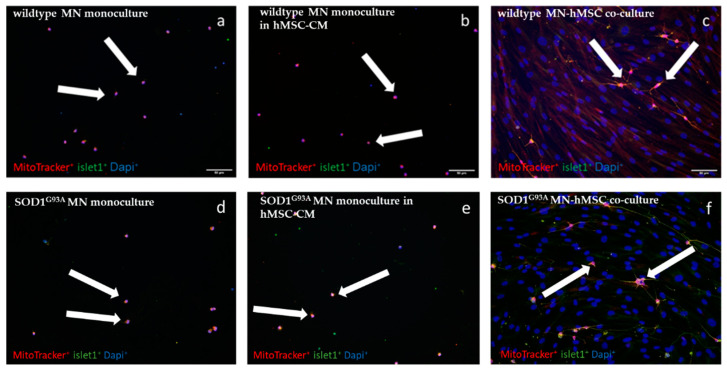
Increased MN number when co-cultured on hMSCs. Representative immunocytochemical stainings of MNs stained for Mitotracker (red), islet1 (green) and DAPI (blue). All images were recorded in a 20× magnification. (**a**) Wildtype MN monocultures. (**b**) Wildtype MNs cultured in hMSC-CM. (**c**) Wildtype MN-hMSC co-cultures. (**d**) SOD1^G93A^ MN monocultures. (**e**) SOD1^G93A^ MNs cultured in hMSC-CM. (**f**) SOD1^G93A^ MN-hMSC co-cultures. All arrows point towards Mitotracker and islet1 double positive MNs.

**Figure 2 biomedicines-10-02916-f002:**
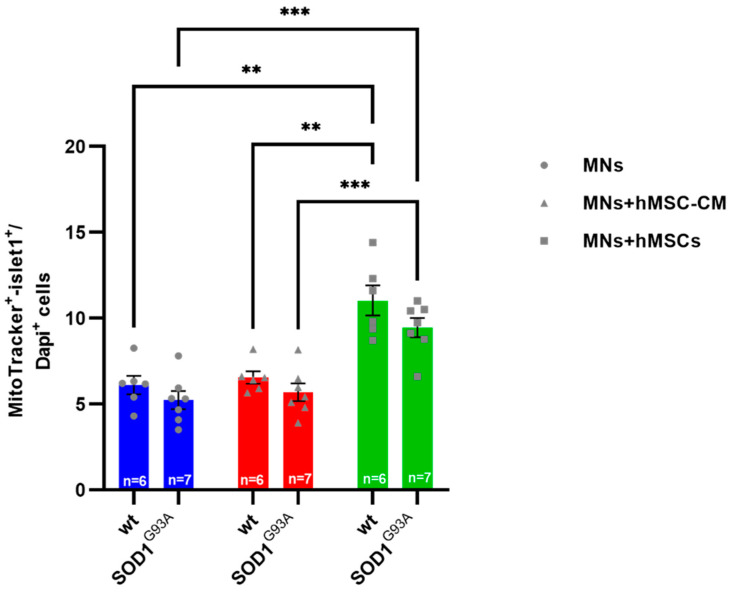
Increased number of MNs when co-cultured on hMSCs. Assessment of MN numbers in primary MN monocultures compared to MN monocultures cultured in hMSC conditioned medium and MN-hMSC co-cultures. 10,000 wildtype or SOD1^G93A^ MN were seeded on 10,000 hMSCs. After 5 DIV cells were fixed and stained for islet1, MitoTracker and DAPI. Double positive (islet1^+^/MitoTracker^+^) cells of 6–7 independent pooled experiments were counted and evaluated. Significant increase of SOD1^G93A^ and wildtype MN number when co-cultured on hMSCs as quantified by immunocytochemistry after 5 DIV (*n* = 6–7). All analyses were performed by 2-way ANOVA: ** *p* < 0.01 and *** *p* < 0.001 and Tukey’s multiple comparisons post-hoc test. Single values are represented as repeated measurements together with mean ± SEM.

**Figure 3 biomedicines-10-02916-f003:**
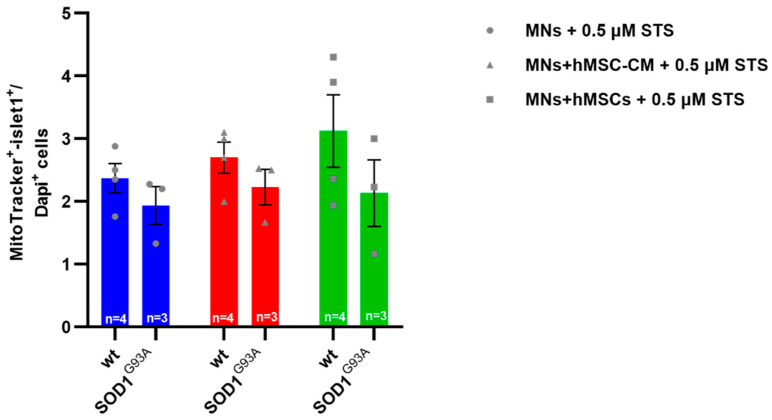
No significant neuroprotective effects of hMSC-CM or hMSC co-culture on STS-exposed wildtype and mutant SOD1^G93A^ MNs. Assessment of MN numbers in MN-hMSC-CM and primary MN-hMSC co-cultures. 10,000 wildtype or SOD1^G93A^ MNs were seeded on 10,000 hMSCs. After 5 DIV cells were exposed to a STS concentration of 0.5 μM for 24 h. After STS exposure, on day 7, medium was removed and cells were fixed and stained for islet1, MitoTracker and DAPI. Double positive (islet1^+^/MitoTracker^+^) cells of 3 independent pooled experiments were counted and evaluated. No change in wildtype or SOD1^G93A^ MN number when cultured in hMSC-CM or co-cultured on hMSCs after 0.5 μM STS exposure as quantified by immunocytochemistry after 7 DIV (*n* = 3). All analyses were performed by 2-way ANOVA and Tukey’s multiple comparisons post-hoc test. Single values are represented as repeated measurements together with mean ± SEM.

**Figure 4 biomedicines-10-02916-f004:**
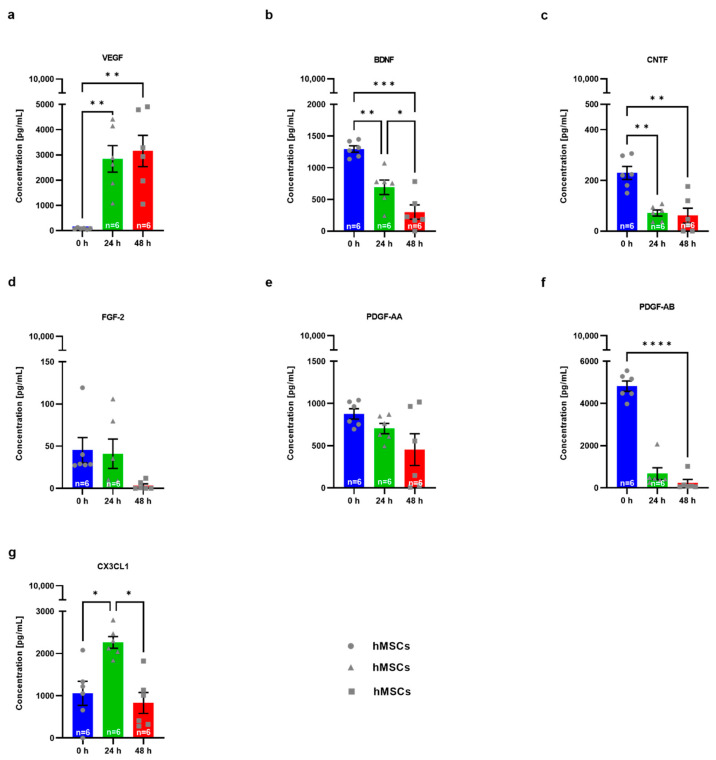
Secretome analysis of protein levels of growth factors in hMSC medium. 100,000 hMSCs were seeded and after 5 min (indicated as timepoint 0), 24 h and 48 h cell supernatant was collected. Supernatants were co-incubated with respective beads, and incubated in assay buffer. After washing mixed biotinylated secondary antibodies were added and incubated, followed by washing. Quantification was performed by adding SA-PE. Individual standard curves and concentrations for each analyte were calculated. Pure hMSC medium was used as a negative control and, thus, subtracted from all measured values. hMSC medium enriched with 1% Penicillin/Streptomycin, heparine 2 IU/mL and 8% human plate lysate was used for baseline measurement and thus subtracted from all measured values. (**a**) VEGF protein levels (*n* = 6). (**b**) BDNF protein levels (*n* = 6). (**c**) CNTF protein levels (*n* = 6). (**d**) FGF-2 protein levels (*n* = 7). (**e**) PDGF-AA protein levels (*n* = 6). (**f**) PDGF-AB protein levels (*n* = 6). (**g**) CX3CL1 protein levels (*n* = 6). Statistical analyses were performed with repeated measurements one-way ANOVA and Tukey’s multiple comparisons post-hoc test: * *p* < 0.05, ** *p* < 0.01, *** *p* < 0.001 and **** *p* < 0.0001. Single values are represented as repeated measurements together with mean ± SEM.

**Figure 5 biomedicines-10-02916-f005:**
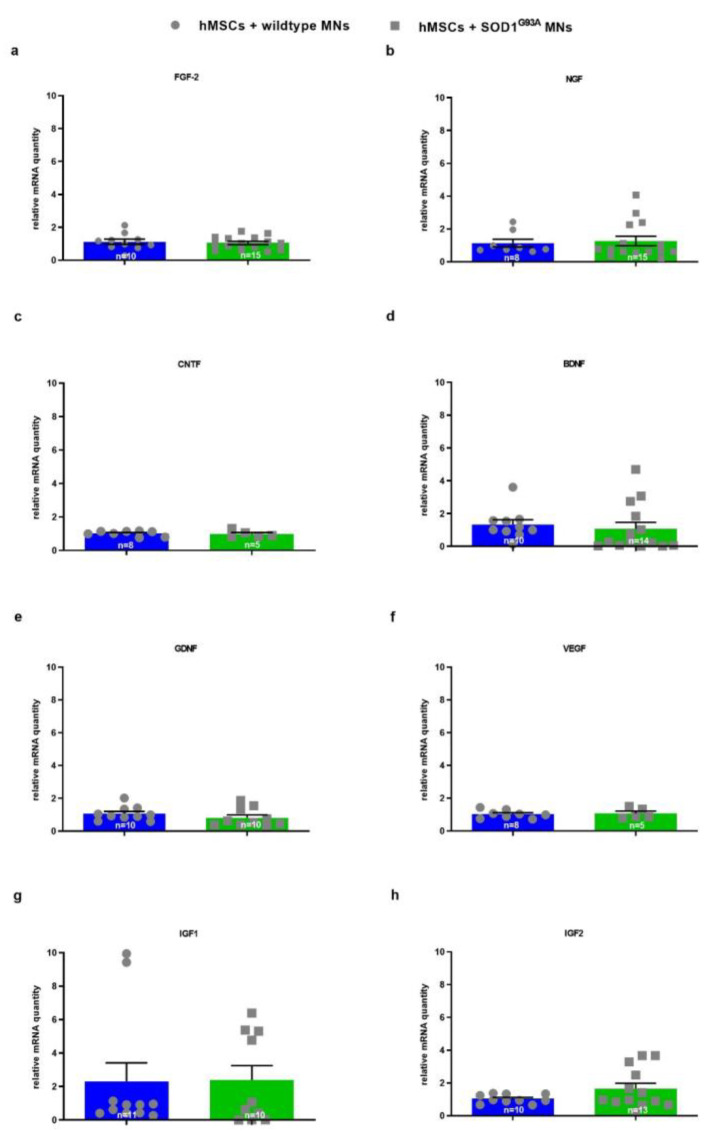
No change in mRNA levels of growth factors in hMSCs when co-cultured with wildtype or SOD1^G93A^ MNs. 10,000 wildtype or SOD1^G93A^ motor neurons were seeded on 10,000 hMSCs. After 5 DIV cells were lysed and mRNA was isolated and transcribed to cDNA. For real-time PCR 5 ng, 20 ng or 50 ng transcribed cDNA per sample were applied, respectively. Quantitative analysis was performed with HPRT1 (for IGF-2), Tubulin (for CNTF and BDNF, respectively), Ppia (for GDNF and IGF1 respectively), b2M (for VEGF, FGF and NGF respectively) as housekeeping genes and expression patterns were analysed by comparative Ct method (2^−ΔΔCt^). (**a**) FGF-2 mRNA levels (*n* = 10–15). (**b**) NGF mRNA levels (*n* = 8–15). (**c**) CNTF mRNA levels (*n* = 8–10). (**d**) BDNF mRNA levels (*n* = 10–14). (**e**) GDNF mRNA levels (*n* = 10). (**f**) VEGF mRNA levels (*n* = 5–8). (**g**) IGF1 mRNA levels (*n* = 10–11). (**h**) IGF2 mRNA levels (*n* = 10–13). Statistical analyses were performed with unpaired *t* test. Single values are represented as repeated measurements together with mean ± SEM.

**Figure 6 biomedicines-10-02916-f006:**
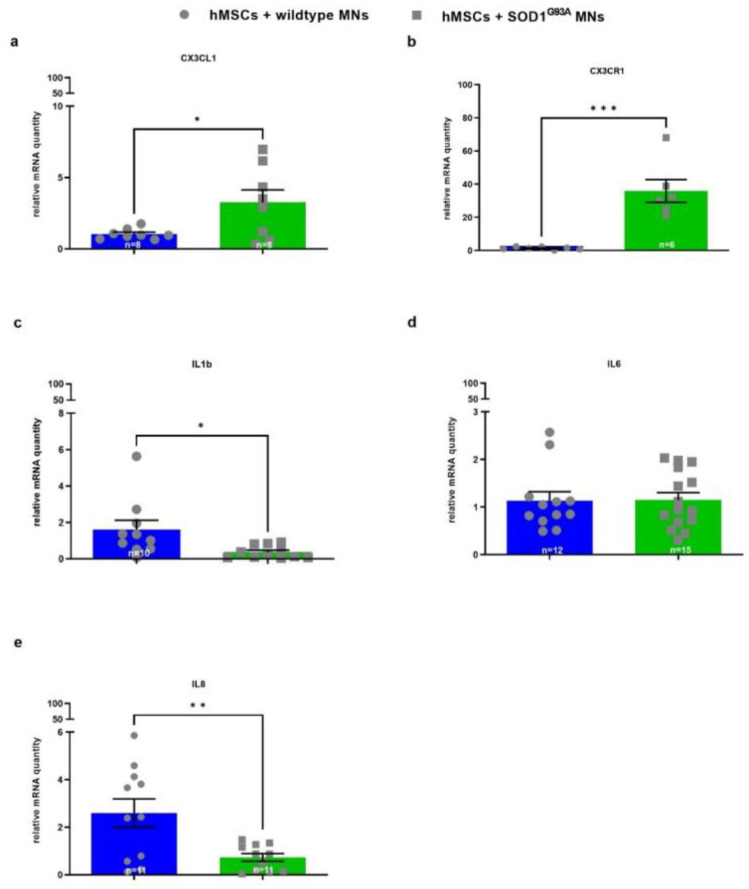
Altered mRNA levels of inflammatory cytokines/chemokines (CX3CL1, CX3CR1, IL1b and IL8) in hMSCs when co-cultured with wildtype or SOD1^G93A^ MNs. 10,000 wildtype or SOD1^G93A^ motor neurons were seeded on 10,000 hMSCs. After 5 DIV cells were lysed and mRNA was isolated and transcribed to cDNA. For real-time PCR 5 ng or 50 ng transcribed cDNA per sample were applied, respectively. Quantitative analysis was performed with Tubulin (for IL8), PPIA (for CX3CL1, CX3CR1 and IL1b, respectively) and b2M (for IL6) as housekeeping genes and expression patterns were analysed by comparative Ct method (2^−ΔΔCt^). (**a**) CX3CL1 mRNA levels (*n* = 8). (**b**) CX3CR1 mRNA levels (*n* = 6–7). (**c**) IL1b mRNA levels (*n* = 10–11). (**d**) IL6 mRNA levels (*n* = 12–15). (**e**) IL8 mRNA levels (*n* = 11). Statistical analyses were performed with unpaired *t* test: * *p* < 0.05, ** *p* < 0.01 and *** *p* < 0.001. Single values are represented as repeated measurements together with mean ± SEM.

**Figure 7 biomedicines-10-02916-f007:**
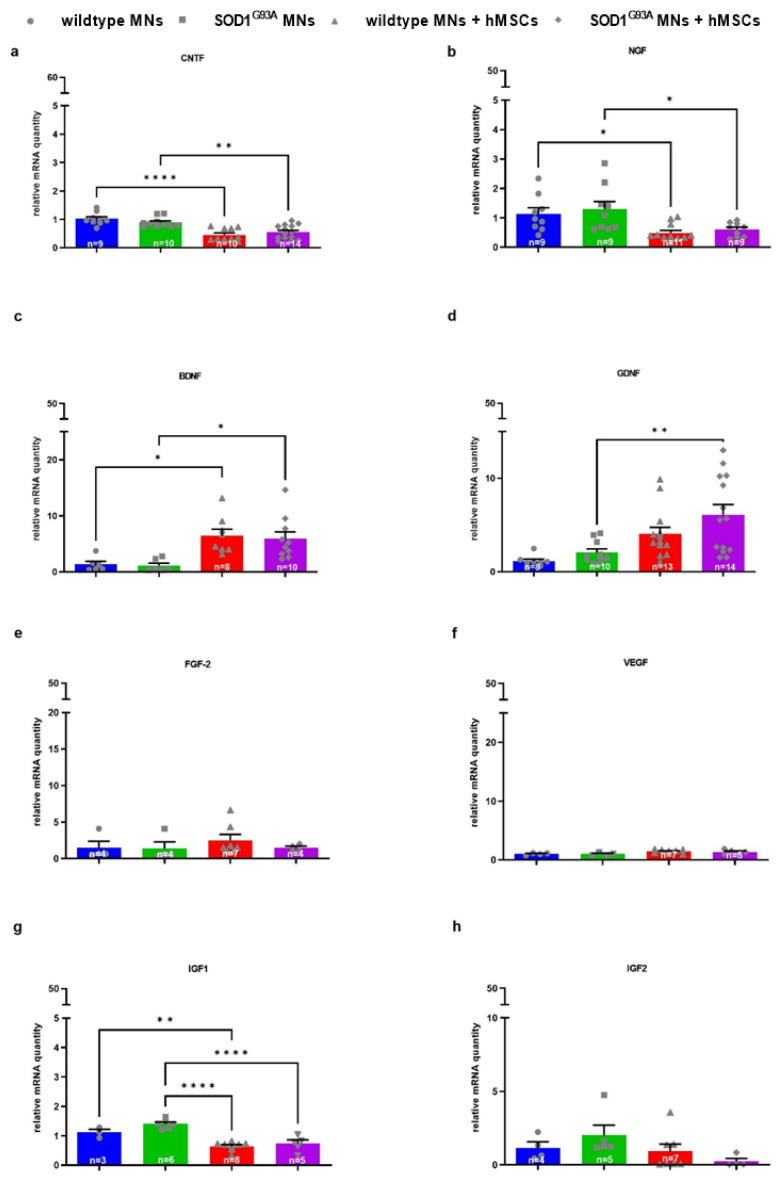
Altered mRNA levels of growth factors (CNTF, NGF, BDNF, GDNF and IGF-1) in MNs co-cultured with hMSCs. 10,000 wildtype or SOD1^G93A^ MN were seeded on 10,000 hMSCs. After 5 DIV cells were lysed and mRNA was isolated and transcribed to cDNA. Quantitative analysis was performed with HPRT1 (for CNTF, IGF1, IGF2, respectively), Act (for VEGF, BDNF, GDNF, respectively), PPIA (for FGF-2) and GAPDH (for NGF) as housekeeping genes and expression patterns were analysed by comparative Ct method (2^−ΔΔCt^). (**a**) CNTF mRNA levels (*n* = 9–14). (**b**) NGF mRNA levels (*n* = 9–11). (**c**) BDNF mRNA levels (*n* = 6–10). (**d**) GDNF mRNA levels (*n* = 8–14). (**e**) FGF-2 mRNA levels (*n* = 4–7). (**f**) VEGF mRNA levels (*n* = 4–7). (**g**) IGF1 mRNA levels (*n* = 3–8). (**h**) IGF2 mRNA levels (*n* = 4–7). Statistical analyses were performed with ordinary one-way ANOVA and Tukey’s multiple comparisons post-hoc test: * *p* < 0.05, ** *p* < 0.01 and **** *p* < 0.0001. Single values are represented as repeated measurements together with mean ± SEM. Statistic analysis between monocultures and co-cultures of different genotypes are not shown.

**Figure 8 biomedicines-10-02916-f008:**
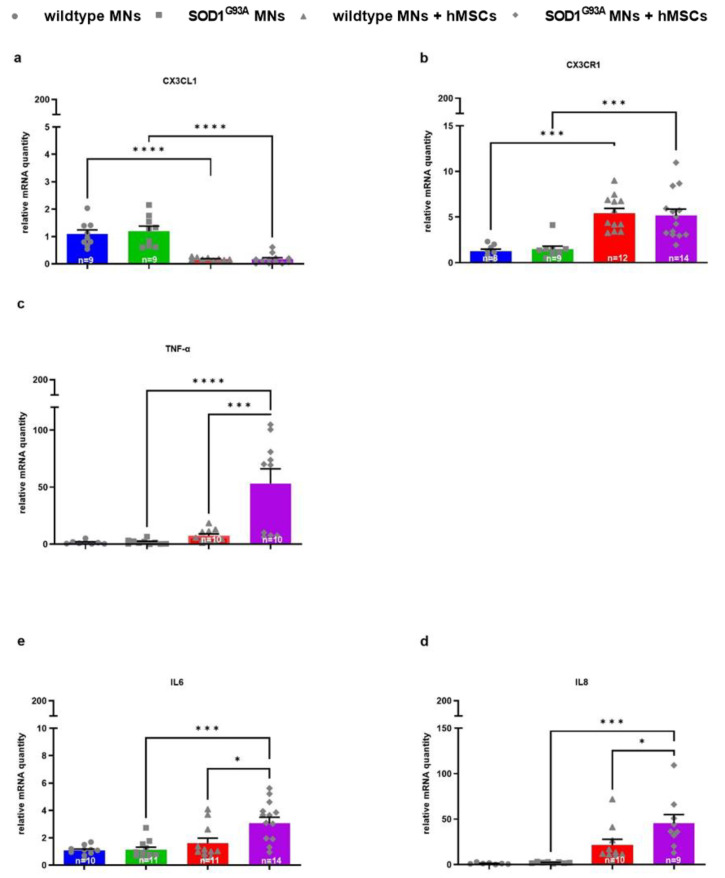
Altered mRNA levels of cytokines/chemokines (CX3CL1, CX3CR1, TNF-α, IL6 and IL8) in MNs co-cultured with hMSCs. 10,000 wildtype or SOD1^G93A^ MN were seeded on 10,000 hMSCs. After 5 DIV cells were lysed and mRNA was isolated and transcribed to cDNA. Quantitative analysis was performed with HPRT1 (for CX3CL1, IL6, IL8, respectively), PPIA (for CX3CR1) and GAPDH (for TNFα) as housekeeping genes and expression patterns were analysed by comparative Ct method (2^−ΔΔCt^). (**a**) CX3CL1 mRNA levels (*n* = 9–12). (**b**) CX3CR1 mRNA levels (*n* = 8–14). (**c**) TNFα mRNA levels (*n* = 8–10). (**d**) IL8 mRNA levels (*n* = 7–9). (**e**) IL6 mRNA levels (*n* = 10–14). Statistical analyses were performed with ordinary one-way ANOVA and Tukey’s multiple comparisons post-hoc test: * *p* < 0.05, *** *p* < 0.01 and **** *p* < 0.0001. Single values are represented as repeated measurements together with mean ± SEM. Statistic analysis between monocultures and co-cultures of different genotypes are not shown.

## Data Availability

The original contributions presented in the study are included in the article/Appendix A, further inquiries can be directed to the corresponding author.

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
