# Peer review of "Altered Immunomodulatory Responses in the CX3CL1/CX3CR1 Axis Mediated by hMSCs in an Early In Vitro SOD1^G93A^ Model of ALS"

_biomedicines, 2022, doi:10.3390/biomedicines10112916_

Round 1
Reviewer 1 Report (Previous Reviewer 1)
Although I appreciate the response to my previous concerns about the manuscript and the changes made by the authors, this new version of the manuscript does not fulfill my doubts about the validity and the overall message of the work. Here, in details my main concerns.
The authors used primary cultures instead than NSC-34 cells (that are not glioblastoma cells as specified in lane 96, but they derive from neuroblastoma cells and mouse spinal cord MNs); however, they weren’t able to confirm the beneficial effect of hMSCs and/or CM in this model, as instead previously observed in the previous work by Sun et al. Authors show only a “cell count” experiment of mono- and co-cultures from which emerges an increased cell number, without investigating the cause. Are hMSCs and/or CM able to induce proliferation or they counteract apoptotic process in SOD1 G93A cells? A similar results is showed in Fig. 3, where a proportional cell increasement is showed. Therefore, I cannot agree with statements of paragraph 3.1.2, since the slight increase of cell number appears likely the same effect shown in Fig. 2 and not a neuroprotective effect against STS treatment. Overall, these experiments don’t add much and, rather, are opposed to previous findings.
The part investigating the secretome is interesting, albeit some inconsistencies emerged. In Fig. 4, authors show the protein concentration of growth factors secreted by hMSCs; thus, we can assume that same proteins were present in co-cultures and in the CM experiments. Despite this, they seems to have a very limited effect on primary MNs (promoting a slight increasement of cell number but not counteracting STS) while they possibly exert a positive function in astrocyte and NSC-34 cells as previously shown in Sun et al work. Also, it is not completely clear the rationale about investigating the influences of MNs (expressing WT or SOD1 G93A) on hMSCs if the last should be used for supporting ALS MNs. Overall these data do not add much.
I found also some technical problems. The quality and magnification of the microscopy images in Fig. 1 is not appropriate. It is, indeed, very hard to distinguish the colors (i.e., the green of islet1 marker, etc). Furthermore, where is the DAPI signal in a, b, d and e? Also, the image is very hard to interpreted. Authors should add the information about samples (as they did in the previous paper), showing on the right “SOD1 WT” or “G93A” MNs, and on the top if they are mono-, co-cultures ot CM treated. In Fig. 2, it is not clear how the quantification was performed, neither how much cells were counted for each count; authors should clearly write this information at least in the figure legend. Also, statistical data about comparison between SOD1 WT and G93A MNs should be showed (it is a control).
Other minor correction should be made:
Lane 54: the extensive name for SOD1 is “Cu/Zn Superoxide Dismutase”.
Lane 79: hMSCs is not defined (only in the abstract).
Lane 96: NSC34 cells are not from glioblastoma cells.
Lane 97: MCS-CM is not defined. Furthermore, it is not clear what is CX3CL1/CX3CR1 axis why is important for the study. Authors should properly introduce it.
Paragraph 2.2: mice genotype is missing.
Paragraph 2.3: this paragraph appears to long and should be divided in different sub-paragraphs (Isolation, Differentiation, etc…) Supplementary table must be moved in a separate file; some technical reference about protocols should be added.
Lane 281: a reference about the quantification method is missing.
Lane 340: does some reference exist about the use of STS as stress factor in ALS?
Lane 371: the neuroprotection is not showed, so in this case the sentence is not correct.
For all these reasons, the manuscript does not deserve the publication on Biomedicines at the stage.
Author Response
Please see the attachment.

Reviewer 2 Report (New Reviewer)
Amyotrophic lateral sclerosis (ALS) is a progressive neurodegenerative disease that results in respiratory failure and death, usually within a few years of diagnosis. In spite of the intensive research performed for decades we are far from effective treatment. Current therapies for ALS are sparse, and available therapeutic drugs—riluzole and only in some countries edaravone—offer only modest clinical benefit. Searching for a treatment for this devastating disease is a challenge of major importance. Over the past years, cell therapy has been explored as an alternative to pharmacological treatment in ALS in order to prevent motor neuron (MN) degeneration by providing a trophic environment. As the authors mentioned human mesenchymal stem/stromal cells (hMSC) can prolong MN survival by secretion of growth factors and modulation of cytokines/chemokines. The introduction of the article provides sufficient background for the topic.
The authors investigated the effects of hMSCs on SOD1G93A transgenic primary motor neurons. The research design is appropriate. The references contain the main important publications in the investigated field. The included tables and figures clearly demonstrate the obtained results. The authors concluded that hMSCs may mediate anti-inflammatory responses through the CX3CL1/CX3CR1 axis. The obtained results confirm that hMSCs provide trophic support to MNs by growth factor gene regulation.
Author Response
Please see the attachment.

This manuscript is a resubmission of an earlier submission. The following is a list of the peer review reports and author responses from that submission.
Round 1
Reviewer 1 Report
The work by Sarikidi et al characterizes the beneficial effects of hMSCs when the cells are co-cultured with MNs expressing SOD1 WT and G93A, the last a widely used model of ALS. Substantially, authors repeat the experiments already performed in their previous paper by Sun et al (Plos One, 2013) in which they investigated the effect of MSC co-cultures and condition media onto three different cell types, including astrocytes and NSC34 cells. In the current paper, most of the presented results overlap with previous data or are in contrast with them. For instance, the impact of MSC co-culture on the transcriptional activity of MNs appear very similar (if not identical) to that found with the utilization of CM, and STS results appear to debunk previous findings. To me, it appears that the only new result is that co-culture increase the number of MNs; the rest is a list of preliminary/control experiments that were not included in the first manuscript for some reason. Moreover, the manuscript is confusing, not properly written, and this does not help to understand what is the point. In my opinion, at this stage the manuscript does not merit the publication.
Reviewer 2 Report
In the Graphical Art, SOD1G93A was incorrectly spelled as SOD1G39A.
The way experiments were described, it is not clear if MSCs were co-cultured with wildtype MNs, or with SOD1G93A MNs. As a consequence, it is not clear if the upregulation of the growth factors and the anti-inflammatory molecules was observed only when co-cultured with ALS MNs, or with healthy MNs as well. In other words, is this effect coming from hMSCs, or from the diseased ALS MNs? The authors should clarify.
The percentage of cells with the expression of CD73, CD90 and CD105 in the population of BM-MSCs as determined by FACS should be indicated.
The MNs were derived from D12.5 embryos. Since ALS is a delayed-onset disease, not manifested until young age, these SOD1G93A MNs may be considered “normal.” The authors should discuss the possible implications.